# Soil Macrofauna: A key Factor for Increasing Soil Fertility and Promoting Sustainable Soil Use in Fruit Orchard Agrosystems

**Adriano Sofo [1],\*** , **Alba Nicoletta Mininni [1]** and **Patrizia Ricciuti [2]**

[1] Department of European and Mediterranean Cultures: Architecture, Environment and Cultural Heritage (DiCEM), University of Basilicata, Via Lanera, 20–75100 Matera, Italy; alba.mininni@unibas.it
[2] Department of Soil, Plant and Food Sciences (DiSSPA), University of Bari 'Aldo Moro', Via Amendola, 165–70126 Bari, Italy; patrizia.ricciuti@uniba.it
\* Correspondence: adriano.sofo@unibas.it

**Abstract:** Soils and crops in orchard agrosystems are particularly vulnerable to climate change and environmental stresses. In many orchard soils, soil biodiversity and the ecosystem services it provides are under threat from a range of natural and manmade drivers. In this scenario, sustainable soil use aimed at increasing soil organic matter (SOM) and SOM-related benefits, in terms of soil quality and fertility, plays a crucial role. The role of soil macrofaunal organisms as colonizers, comminutors and engineers within soils, together with their interactions with microorganisms, can contribute to the long-term sustainability of orchard soils. Indeed, the continuous physical and chemical action of soil fauna significantly affects SOM levels. This review paper is focused on the most advanced and updated research on this argument. The analysis of the literature highlighted that a significant part of soil quality and fertility in sustainably-managed fruit orchard agrosystems is due to the action of soil macrofauna, together with its interaction with decomposing microorganisms. From the general analysis of the data obtained, it emerged that the role of soil macrofauna in orchards agrosystems should be seriously taken into account in land management strategies, focusing not exclusively on fruit yield and quality, but also on soil fertility restoration.

**Keywords:** bioturbation; ecosystem engineers; orchards; soil biodiversity and fertility; soil macrofauna; soil organic matter

## 1. Introduction

From an ecological point of view, the soil is a dynamic habitat for an enormous variety of life-forms [1]. The soil–food web consists of an extensive web of biotic interactions that jointly determine many soil processes and in turn ecosystem functions [2].

In many soils covered by orchards, soil biodiversity (including bacteria, archaea, actinomycetes, fungi, algae, protists, invertebrates, etc.) and the ecosystem services it provides are under threat from a range of natural and manmade drivers, including human management of agricultural and other soils [3]. In many parts of the world, fruit groves are endangered by an increasing water shortage often due to changes in rainfall frequency and distribution, and rise of soil aridity and desertification, with resulting critical and low levels of soil organic matter (SOM) and contents of macro- and micronutrients [4]. This insufficient return of SOM to orchard soils has led to severe degradation of the productive capacity of naturally poor soils [5]. Furthermore, the adoption of non-sustainable agricultural practices reduces soil biodiversity due to the host specificity of many of the soil bacteria and fungi and the higher trophic level organisms (micro- and mesofauna) that they attract [6,7]. Examples include negative effects of

tillage, mineral fertilizers and pesticides on the genetic, functional and metabolic diversity of soil microorganisms [8–10].

While the importance of microorganisms (particularly of bacteria and fungi) in orchard agrosystems has been extensively and recently highlighted [4,5,9–11], the role of soil fauna—and particularly of macrofauna—to ecosystem services has been often overlooked [12,13], while it should seriously be taken into account in land management strategies that are not only focused on fruit production. Indeed, in a holistic and realistic vision of orchard agrosystems, most soil processes and biogeochemical cycles are regulated not only by soil microbial communities but by the whole soil–food web [13]. It is known that animals living in the soil can be viewed as facilitators of bacterial and fungal activity and diversity. Thus, the interactions between belowground microbes, microfauna (e.g., protists, small nematodes; rotifers, tardigrades; body width < 100 μm), mesofauna (e.g., springtails, detritivore, and predatory mites, proturans, symphylans; body width between 100 μm and 2 mm), and macrofauna (e.g., earthworms, isopods, myriapods, collembola, and insects such as ants, carabids, termites, cicadas and many species of caterpillars; body width between 2 mm and 20 mm) are of key importance. In particular, macrofaunal organisms are ecosystem engineers able to ameliorate soil physical structure, mineral and organic matter composition, and hydrology, influencing nutrient and energy flow and forming a connection between the food chains of the foliage and the soil [14].

Among the drivers of soil biological fertility linked to soil management in orchard soils, many can be determined, but soil macrofauna abundance has been scarcely explored. A wide range of studies in orchard soils pointed out that this factor is strongly and positively related to the application of sustainable and conservation agricultural practices, such as no- or minimum soil tillage, use of organic fertilizers (e.g., compost addition), guided irrigation, and recycling of polygenic carbon sources (such as cover crops and pruning material), applied alone or in combination [4,9,10]. Abundant soil macrofauna can contribute to the creation of heterogeneous ecological niches and favorable microclimates that in turn increase genetic, functional and metabolic soil diversity of the soil microflora involved in nutrient cycles, so enhancing soil quality and fertility. Furthermore, a great part of the soil quality and fertility in an orchard agrosystem is due to the bioturbation activity of soil macrofauna (the transport of particles from lower horizons to the surface that aids in mixing the organic and mineral fractions of the soil) [14,15].

On this basis, in this review paper, we focused our attention on the most advanced and updated research on the role of soil macrofauna in increasing soil fertility and, consequently, promoting sustainable soil use in fruit orchard agrosystems. We start with the role of organic matter, soil macrofauna and its changes related to soil management (sustainable/integrate/organic vs. conventional), then move onto the specific soil macrofauna of fruit orchards (with a particular emphasis on earthworms, ants, termites and beetles) and its numerous advantages and benefits on the quality and fertility of orchard soils, also considering the controversial and potentially negative effects. Indeed, all parts of this complex scenario should be taken into account for defining the biological quality and fertility of orchard soils and are critical for understanding how soil fauna, microbes, and plants interact and respond to the multiple global changes expected in the future.

## 2. The Scenario: Soil Organic Matter

Organic matter is the center of nearly all life activities in the soil including that of the microflora, the fauna, and the root systems of higher plants. Soil organic matter (SOM), especially its stabilized fraction (humus) (Figure 1), plays a crucial role in soil physical, chemical and biological fertility [16]. The amount and types of SOM are principally determined by the continuous physical and chemical action of soil organisms. Soil fauna and microbes are crucial for shredding, transformation and decomposition of SOM [1,2]. For this reason, studies focused on the understanding of soil microorganisms–SOM and macrofauna–SOM interactions are particularly relevant.

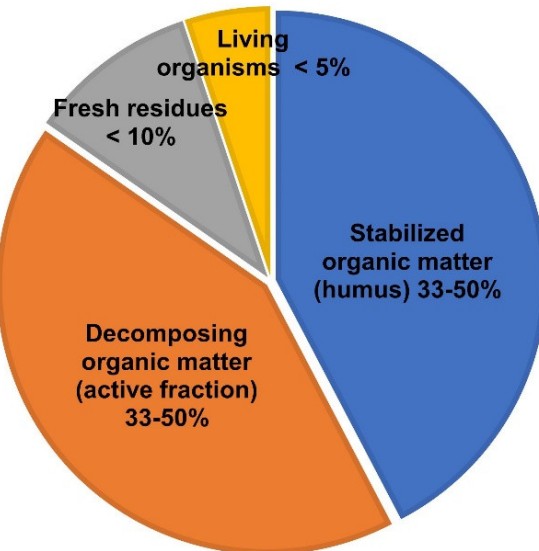

**Figure 1.** Soil organic matter is carbon-based substances in soil, including leaves, roots and living organisms such as earthworms [17].

In particular, orchard soils are vulnerable to climate change and environmental stress, and they will continue to be, more and more in the future [3]. Many environments are endangered by increasing water shortage, often due to changes in rainfall frequency, and the rise of soil aridity and desertification, eventually resulting in deteriorated soil structure and critically low levels of SOM, macro- and micronutrients [10,15,18]. The frequent and strong soil tillage, typical of the intensive tree growing, significantly affects the stability of soil microaggregates, which have a key role in SOM stabilization and support long term carbon sequestration, being more stable than macroaggregates [14,19,20]. The massive adoption of inorganic fertilizers can immediately increase fruit yields but, when used alone, they may not provide enough crop residue to increase SOM [5,21]. This triggers a vicious detrimental circle which ultimately leads an increase in the use of mineral fertilizers and pesticides, and continuous and strong soil tillage, again increasing SOM loss [22,23].

Fertile soils are rich in SOM built of carbon that living plants has removed from the atmosphere through photosynthesis [24,25]. Unfortunately, soils can release carbon, too. The frequent tillage and heavy fertilizer use characteristic of modern conventional agriculture have accelerated SOM degradation, releasing more carbon into the atmosphere. The new IPCC report [16] concludes that, globally, cropland soils have lost 20 to 60 percent of their original SOM content. Additionally, increasing SOM level enhances soil ability to hold water, also reducing reduce soil erosion. This stored water can better sustain tree crops, especially during drought-stressed years [10]. In addition to benefiting the climate, less fertilizer use can decrease off-farm water pollution, reducing economic costs for farmers [22]. Land management choices also affect the amount of carbon stored in trees, plants and soil [26]. The last IPCC report [16] estimates that serious changes in forestry and agriculture, to curtail deforestation and improve soil management, could reduce global emissions by 5 to 20 percent.

The different chemical and physical fractions of SOM affect the structure of soil macrofaunal communities and the functioning of soil–food webs [22–24]. In particular, particulate organic matter (POM; between 0.053 mm and 2 mm in size) and mineral-associated organic matter (MOM). The decomposition of POM, regulated by the C/N ratio, provides energy and nutrients for soil organisms, including macrofauna, and plants. Moreover, POM plays a role in the formation of soil aggregates, ultimately improving soil structure [5,21]. The POM fraction rapidly decreases if soils are subjected to repeated tillage or other soil disturbance, being highly sensitive to soil management for this reason. Thus, POM is used as an indicator to measure soil quality and, generally, if the POM fraction in the soil is high, macrofaunal organisms are abundant and have a higher diversity at both

specific and community levels [10,18]. Conversely, the accumulation of MOM fractions, the formation of which involves various mechanisms and processes, is little influenced by litter quality, less affected by the agronomic practices adopted, and has a negligible influence on the dynamics of soil macrofauna, compared to POM [5,23,26].

## 3. The Actors: Soil Biota in Agrosystems

Organic matter inputs to soil are critical for supporting diverse and active soil organisms, responsible for residue decomposition, carbon turnover, nutrient cycling, and other functions, such as disease and pest suppression [27]. Soils host a wide diversity of organisms that play fundamental roles in driving many ecosystem processes on which the functioning of terrestrial ecosystems depend [28–32]. One gram of soil contains up to one billion bacterial cells, up to 200 m of fungal hyphae, and a high number of invertebrates, that are all part of a complex and interconnected food web (Figure 2). These organisms coexist and interact with viruses of unknown impact on susceptibility to disease, nutrient cycles, food web interactions and other key processes [33–35].

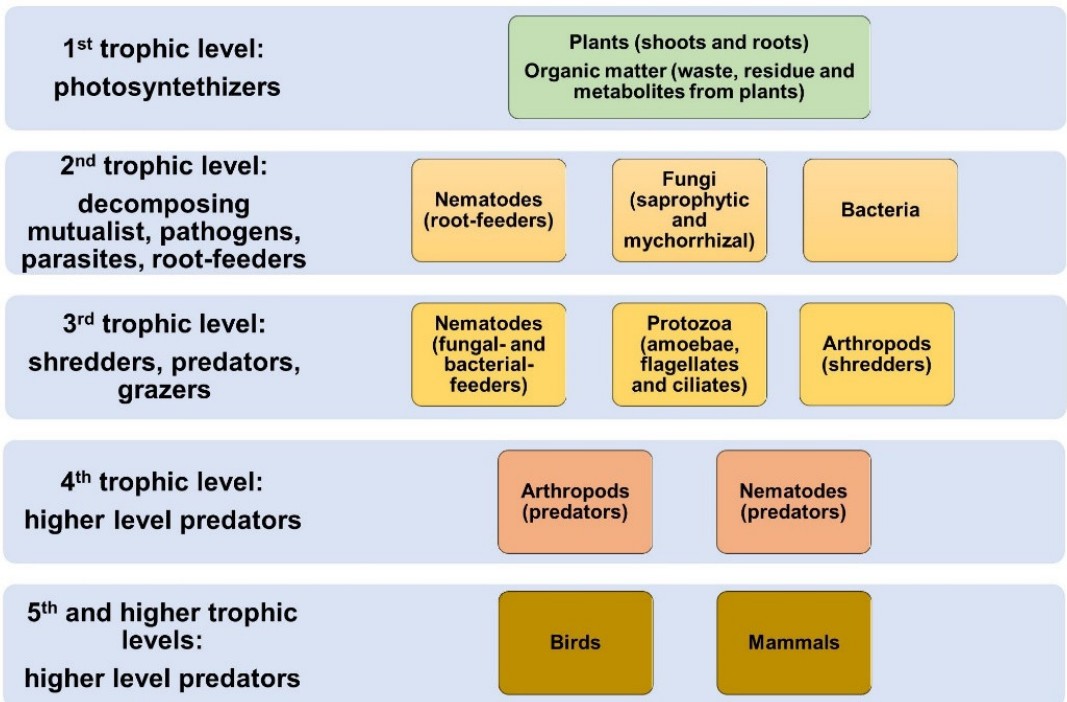

**Figure 2.** The hierarchical structure of the soil–food web.

The health of all multicellular organisms (including plants, animals, and humans) and their surrounding ecosystems are interconnected through a subset of microbial organisms found in the environment, particularly in soil. As such, microbial interactions play a critical role not only in regulating ecological functions and processes but ultimately in determining the health of plants, animals and humans, as components of terrestrial ecosystems [1,32]. While soil biodiversity represents an important biological and genetic resource for biotechnological innovation with benefits to society, it is increasingly threatened by different forms of land degradation [24,25]. Soil biodiversity is vulnerable to many human disturbances, including intensive and non-sustainable agricultural practices, land use, climate change, nitrogen-enrichment, soil pollution, invasive species, and soil sealing [36].

Numerous studies have shown that the conversion of natural lands to agriculture, together with agricultural intensification that enhances SOM depletion, makes the greatest contributions to soil biodiversity loss [9,15,36,37]. The large-scale use of pesticides may also have direct or indirect effects on soil biodiversity, but the lack of data has resulted in contradictory research results [22,38]. Despite

significant data gaps, there is growing evidence that unsustainable agricultural practices not only affect the health and quality of the soils needed to sustain healthy crops and nutrient-rich foods, but they can also have significant impacts on the integrity and resilience of the ecosystem as a whole. Therefore, nature-based solutions are required to facilitate sustainable use and conservation of soils.

It is essential to link biodiversity measures with specific soil functions under particular environmental contexts, particularly in agrosystems [39]. For instance, while some soil functions are driven by a diverse set of organisms that contribute to functional resilience (e.g., decomposition), other soil functions involve a more specific set of organisms (e.g., nitrifiers, bio-control agents) which make them more vulnerable to biodiversity loss. It was found that the simplification of soil community composition can impair multiple ecosystem functions, including plant diversity, decomposition, nutrient retention, and nutrient cycling [33]. Furthermore, a better understanding of the pivotal roles of soil organisms in mediating soil-based ecosystem services, as affected by ecosystem management approaches and practices adapted to socioecological contexts, is also central to guide biodiversity-friendly agricultural intensification trajectories [12,26].

## 4. Soil Macrofauna Abundance and Activity Are Related to Soil Management

The role of macrofauna in soil transport and mixing can be extensive and involve up to several tons per hectare annually of more readily assimilable organic matter. In sustainable agrosystems, earthworms benefit plant metabolism and crop production, cause increased nitrogen-fixing activity, greater amounts of available macro- and micronutrients and polysaccharides, and an enhanced biosynthesis of plant growth regulators [27,40]. In many orchard soils, earthworms are sensitive to the nutrient status and SOM levels and they are abundant under sustainable land management systems, where usually enough litter is available [29,41]. Similar trends may be also found for different soil invertebrates, along with millipedes or beetles, soon disappearing and not replaced beneath intensive agricultural practices, whilst different macrofauna groups, inclusive of ants and termites, tend to be more persistent [42–44].

In most cases, intensive cropping causes a dramatic decrease in taxonomic richness, density, and biomass of soil fauna. When agricultural intensification and traditional, non-sustainable management are adopted, the abundance of the main groups of soil macrofauna decreases with the worsening of soil microclimate, the reduction in litter availability, the physical disruption of habitats, the perturbation induced by soil tillage, and the effects of mineral fertilizers and nontargeted pesticides [45]. In contrast, the spontaneous or cultivated cover crops adopted in many sustainable orchards create a useful vegetal cover, resulting in a more humid soil environment that positively affects the activity of soil organisms. Here, the higher root density causes larger and more abundant soil macropores, which in turn increase water infiltration and aeration, also preventing surface crusting and erosion of topsoil, with positive effects on soil macrofauna and nutrient mineralization [7,45]. In sustainable agrosystems, the improved soil macroporosity due to earthworm activity allows plant roots to go deeper into the soil and favors tree crops that prefer an aerated soil for their development, also reducing the incidence of soil-borne disease. The association of tree crops with spontaneous or cultivated cover crops can present a larger earthworm community and high densities of ants, termites and epigeic litter detritivores, which live and feed on the soil surface. Moreover, soils amended with compost or with other endogenous or exogenous carbon sources (pruning residues, thinned fruits, fallen leaves, mowed weeds, etc.) are more favorable for earthworm development, as there is an improvement in leaflitter quantity and quality, and a reduced physical soil perturbation [6,7].

Sustainably-managed orchards tend to have rather diverse and abundant fauna with density and biomass significantly higher than that of conventionally-managed systems (Figure 3). Aside from earthworms, there is usually an abundant population of nematodes, crickets, grasshoppers, snails, mites, a wide variety of other insects, numerous worms of all kinds. Thus, a combination of conservation or sustainable practices and enhanced residue quality has deep effects on soil macrofauna and ecosystem health [7,29]. Considering the strict relationships between soil macrofauna abundance and biodiversity

on the one hand, and soil management and environmental disturbance on the other hand, soil macrofauna can be a useful tool for monitoring soil status, agriculture sustainability and environmental health [46]. Many groups of soil invertebrates are highly sensitive to environmental disturbance and are positively affected by the abundance of habitats and ecological niches, refugia, reproduction sites and feeding sources typical of sustainable agrosystems and seminatural environments [47]. Finally, macrofauna-based indicators of soil quality, both at community- or taxon-levels, can give useful information for facilitating the adoption of the most suitable and relevant soil management types.

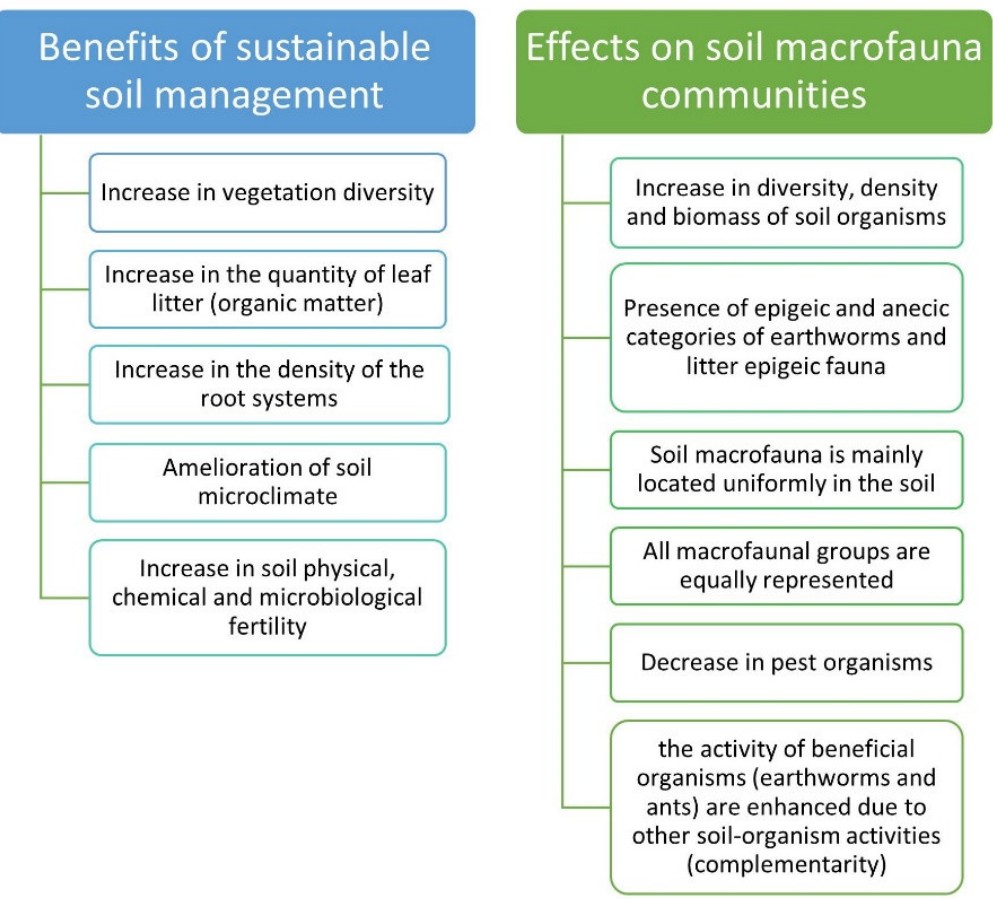

**Figure 3.** Benefits of sustainable soil management and its effect on soil macrofauna.

Another important aspect is the distribution of macrofauna with soil depth. This is relevant for understanding to what degree the type of tillage impacts the soil biota occupying different depths in the soil profile. In most soil types and climates, the biological activity of soil macrofauna is mainly concentrated in the first 20–30 cm of soil [6,29,44]. At the same time, this soil layer is the most disrupted by agronomic activities and where the density of roots (particularly those of herbaceous species) is higher. The life cycle of macrofaunal organisms can be entirely or partially linked to the soil, colonizing soil lates at different depths [6,12,45]. They are conventionally divided into four functional groups: temporary active geophilous, temporary inactive geophilous, periodic geophilous, and geobionts (Figure 4). Inactive geophilous or transitional organisms are in the soil (generally up to a soil depth of 50 cm) to winter or to make the diapause (period of activity interruption induced by temperature and photoperiod), not contributing to soil formation (such as Colorado potato beetle adults, *Leptinotarsa decemlineata*). Active geophilous or temporary or periodic organisms have juvenile stages (larvae and nymphs) of aerial insects living in the topsoil (0–50 cm) (e.g., Diptera belonging to Tipulidae and Bibionidae, Dermaptera, Coleoptera belonging to Scarabeidae). Geophytes or geobionts or permanent organisms spend all their lives in the soil (such as microarthropods), living at deeper soil

layers (up to 3–4 m). According to another classification, temporary active geophilous and temporary inactive geophilous belong to the group of edaphoxenes occasionally present in the edaphic environment and are poorly adapted to hypoqean life; periodic geophilous are included in edaphophilous, preferring the soil, even if they can stay out of it; finally, geobionts are considered edaphobes, completing their cycle in the soil. On this basis, the choice of the soil management type can have marked effects on the composition, variety and abundance of soil macrofauna communities [7,29,41]. Whereas no-tilled soils do not include plowing, soils subjected to minimum tillage can be subjected to "light" management practices (such as harrowing, mechanical weeding) that usually concern the first 0–15 cm of soil, only partially interfering with macrofaunal organisms.

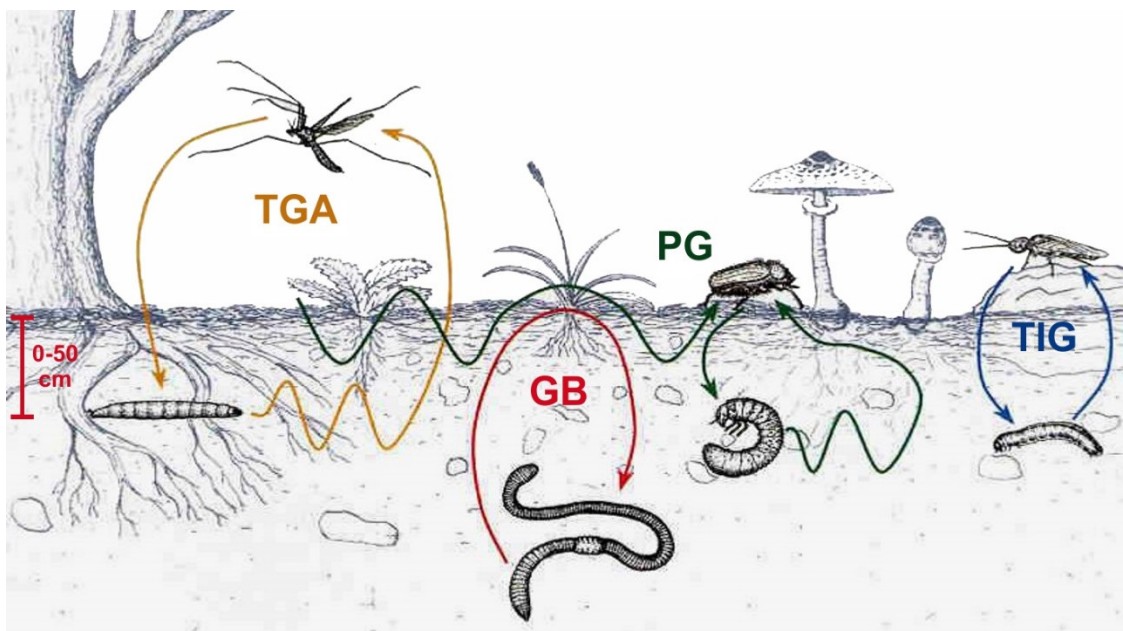

**Figure 4.** The four functional groups of soil macrofauna: temporary active geophilous (TAG), temporary inactive geophilous (TIG), periodic geophilous (PG), and geobionts (GB).

## 5. Soil Macrofauna in Orchard Agrosystems

### 5.1. General Aspects

Perennial fruit crops represent barely 1% of the global agricultural land area, but in many regions, such as in the Mediterranean basin, they cover up to 11% of land and are of great economic importance [48,49]. To assure high yields and fruit quality, fertile soil is fundamental for crops. Soil–root interface is the connection between the plant and the environment [50]. Fertile soil can supply nutrients, to store water from rainfall, to reduce soil erosion, to ensure plant growth and production and to ensure water and nutrient cycles. Soil management is of key importance from an environmental point of view. If it is not well planned and conducted, it can cause decreases in soil organic matter (SOM), mainly due to the mineralization process and consequently increases soil degradation and erosion [51]. Fruit orchards present particular features that could make them interesting from an ecological and environmental point of view. The perennial character of trees, the multi-strata habitat and the plant diversity within the boundaries of fruit orchards make them a unique agricultural system, that could provide multiple ecosystem services [52]. Alternative management of fruit orchards through the application of Good Agricultural Gaps (GAPs) of "Smart Agriculture" can have a key role in maintaining biodiversity and its function [8,10]. Soil fauna abundance and diversity and organic matter content are key indicators for the rate of soil degradation in agrosystems. The greater the degree

of intraspecific or interspecific biodiversity in a functional agricultural ecosystem, the greater will be the tolerance of the latter to perturbation, resulting in an increased resilience [53–55].

Orchard soils are a large reservoir of biodiversity, often little known. In terms of their abundance and their soil-forming roles, earthworms, ants, termites and beetles are the most important macrofauna components of orchard soils. The importance of their activities has caused them to be called "ecosystem engineers", that is, organisms that directly or indirectly modulate the availability of resources to other edaphic species, by causing physical state changes in biotic and abiotic soil materials and by modifying, maintaining and creating habitats [56–61].

### 5.2. Earthworms

In orchard agrosystems, earthworms (phylum Anellida) are often the most important component of the soil macrofauna due to their influence on soil structure and the breakdown of SOM [62]. The most relevant literature on the effects of differential soil management on earthworms dynamics in orchard soils is summarized in Table 1.

In avocado orchard soils, Van Zwieten et al. [63] found that surface-feeding earthworm species horizontally and vertically disseminate microorganisms, spores, pollen and seeds, and can reduce plant pathogens through the digestion of fungal spores. Moreover, in the same orchard, earthworms played an important role in the intimate mixing of organic residues and fine mineral soil particles and the formation of SOM-rich micro-aggregates so they can, therefore, contributing to the physical protection of SOM increasing the soil's potential for carbon sequestration. In a calcareous loam orchard (pear) soil, Jongmans et al. [64] observed that the absence of earthworms led to soil compaction and changes in SOM dynamics (restricted litter incorporation into the mineral soils, less fragmentation of particulate SOM, reduced mixing of SOM with clayey groundmass), with the formation of a litter layer and less physical protection of SOM in soil aggregates. In the studied soil, earthworm species number, abundance and biomass were analyzed to evaluate the impact of the different environmental transformation, such as differential soil management, agricultural practices (intensification of cropping, annual tillage and operations such as fertilization and pesticide use) on the earthworm populations. Higher earthworm numbers were found in no-tilled than in conventionally-tilled orchards, especially for the abundance of epigeic species. In apple orchards, minimum tillage, no-tillage and ridge-tillage tended to reduce the loss of earthworm biomass living on the soil surface [65]. As an example, good earthworm management in orchards can be obtained using a chisel-type tiller that gently mixes the soil avoiding a damaging cutting and slicing action. Paoletti et al. [66] pointed out that both species number and population density of earthworms decreased due to a shift from an organic to a conventional apple orchard. Conventional orchard practices were found to be harmful even to those species of earthworms considered to be more resistant to soil disturbance.

Recently, Castro López et al. [62] compared conventional and organic practices in two biogeographic regions for many orchard types, such as kiwifruit, clementine, grapevine, and olive. Significant reductions in the number of species (*S*), diversity (*H'* index) and mature earthworms have been observed in the drier and warmer climates under conventional practices. The organic olive orchards were dominated by anecic earthworm species, which are very sensitive to ecological disturbances, suggesting that organic practices reduce ecological perturbations compared to conventional ones. Organic management was beneficial for earthworm populations due to the absence of synthetic fertilizers and pesticides, but organic amendments, that provide nutrients for earthworms, proved to be especially useful in the dryer and warm climate. Paoletti et al. [46] studied the response of earthworm communities to soil tillage and fungicide application in different agrosystems, including vineyards and three fruit trees (apple, peach and kiwifruit). Apple and kiwifruit orchards received the highest and the lowest chemical carbon inputs, respectively, whereas peach orchards and vineyards received the medium input. Cultivation operations reduced earthworms in orchards and vineyards in different extent. Epigeic earthworms were severely damaged by tillage, almost disappearing in peach and apple orchards and vineyards. The authors found that copper and zinc contents were

negatively correlated with earthworm species number and biomass, especially for the endogeic forms. In particular, kiwifruit receiving the lowest chemical input, showed the highest number of species.

High concentrations of copper can be found in many orchards treated by fungicides. Copper is an essential element required by all organisms, however elevated concentrations of copper in soils are toxic and may result in reduced biological activity and subsequent loss of fertility [63]. Treatments of orchards with copper sulfate strongly affects earthworms in terms of both biomass and species population response [67]. Moreover, most endogeic species disappear in copper-contaminated soils [68]. In an avocado orchard, Van Zwieten et al. [63] examined different degrees of exposure to fungicides (an orchard, a less contaminated site within the orchard and a nearby control site), noting a lower earthworm density in orchard soils with a history of Cu-fungicide use. Pesticides usually reach soil as mixtures of several products, especially in orchards, and they have the highest effects on the earthworms feeding at the soil surface. In orchards, slurries and sewage are sometimes applied and the resulting soil toxicity deriving from the introduction of contaminants, such as heavy metals and polychlorinated biphenyls, can be monitored by earthworm numbers and dynamic [69,70]. In the study of Reinecke et al. [71], the impact of organophosphate pesticides in a plum orchard resulted in higher densities of earthworms in the pasture area, where the soil was the same of the plum orchard but less disturbed and with a lower pesticide concentration. The authors highlighted that earthworms are very sensitive to soil moisture, so a higher density of worms is expected in the orchard than in the pasture, as the fruit trees are irrigated, but higher pesticide concentration was the limiting factor.

**Table 1.** Effects of differential agricultural management on earthworm dynamics in orchard soils.

| Authors | Year | Fruit Orchard | Agricultural Management | Results | | Reference |
|---|---|---|---|---|---|---|
| Haynes | 1980 1981 | Apple | Retention and removal of grass (either by cultivation or by use of herbicides) | a) | Elimination of the grass produced an unfavorable habitat for earthworm populations | [72] |
| | | | | b) | Cultivation markedly reduced earthworm density (mechanical disturbance) compared to herbicides treatments | |
| | | | | c) | The herbicide treatments resulted in compaction of soil surface | |
| Pizl | 1992 | Apple | Intensively managed soils (trafficked and untrafficked plots) | a) | Decreases of population density and biomass of earthworms living in the upper layers (especially juvenile worms and cocoons) | [73] |
| | | | | b) | The susceptibility of soil to compaction was increased under the spray regime | |
| Iglesias Briones et al. | 2011 | Kiwifruit | Conventional practices with the introduction of anecic earthworm species in combination with nitrogen applications, such as cow manure or planting N-fixing legume | | Only manure applications promoted: | [74] |
| | | | | a) | earthworm activities and biomass | |
| | | | | b) | C and N mineralization and nutrient availability | |
| Lardo et al. | 2012 | Vineyards | Grass cover, chemical weeding or tillage | | Lowest values of earthworm density and biomass, and selective action towards earthworm categories in tillage treatments | [75] |
| Goh et al. | 2012 | Apple | a) Organic, conventional, integrated grassed down production systems  b) Commercial orchard (organic and conventional) | | In both cases (a,b), marked increases in the total number of epigeic earthworm species, fresh biomass and species composition occurred in organic systems, related to high SOM input through grass covering and to the absence of biocides | [76] |
| Lardo et al. | 2015 | Peach | Commercial orchard with a clay loam managed by fertigation | | Abundance and biomass of endogeic and anecic earthworms were significantly higher in inter-rows covered by permanent spontaneous grass cover (no-tillage, no mineral fertilization) than in rows weeded with glyphosate | [77] |

*5.3. Ants*

Formicidae (order Imenoptera) is probably the most significant family of soil insects, due to the very large influence they have on soil structure. Ants are numerous, diverse, and widely distributed from arctic to tropical ecosystems and well organized in a social structure. They are one of the dominant groups of the planet's animal biomass, accounting for around 10% [78]. Ants are major predators of small invertebrates, and they provide several ecosystem services in agrosystems, such as plant pollination, soil bioturbation, bioindication, and the regulation of crop-damaging insects. The multiplicity of their roles is linked to the great diversity of their group compared to other animal taxa [79]. Over recent decades, there have been numerous studies in ant ecology related to their interactions with tree cropping systems and different management, in order to better understand the roles played by ants in orchards as functional elements. The most relevant literature on the effects of differential soil management on ant dynamics in orchard soils is summarized in Table 2.

The galleries mined by ants when building nests, help to aerate the soil and increase soil porosity, which influences water infiltration and circulation and reduces the potential risks of erosion. Also, ants improve the availability of resources for microorganisms and plants [80]. Soil physical structures are also created from the mineral soil around the root: stable aggregates and pores that improve the circulation of air and water to and within deeper soil horizons, facilitating greater root penetration. The immediate root–soil interface is a zone of high porosity, which is very important for several key rhizosphere processes occurring at this scale, including water and nutrient uptake and gaseous diffusion [50]. These effects highlight the importance of ant activity also in agrosystems, in which production sustainability involves the stability of ecological processes linked to soil quality [79].

**Table 2.** Effects of differential agricultural management on ant dynamics in orchard soils.

| Authors | Year | Fruit Orchard | Agricultural Management | | Results | Reference |
|---|---|---|---|---|---|---|
| Cerdà et al. | 2009 | Citrus | Organic orchards | a) | The application of organic amendments to soil promoted high water infiltration rates and ant activity had little effect on these rates | [82] |
| | | | | b) | The ant activity had little effect on soil erosion rates under organic management (with cover crops) | |
| Farji-Brener and Tadey | 2009 | Ecosystem soil | Organically-managed soils | | The building, enlargement, and maintenance of nests ants affected soil structure, porosity and density | [43] |
| Cerdà and Jurgensen | 2011 | Citrus | Intensively-managed orchard | a) | The ants increased water infiltration rates by forming soil macropores during nest construction | [81] |
| | | | | b) | The ant nests in orchard soil decreased water losses by the infiltration of surface runoff into nest macropores | |
| Farji-Brener and Werenkraut | 2017 | Ecosystem soil | Organically- managed soils | a) | The plants inhabiting nest areas showed greater abundance, growth rate, foliar and root biomass, and reproduction rate | [44] |
| | | | | b) | Leaf-cutting ants improved soil fertility and vegetation patterns | |

Ants are widely found in orchard soils of temperate and semi-arid areas, where they can play a key role in the erosion processes on agricultural land by modifying soil properties and increasing macropore flow. Some studies have revealed that they increase water infiltration rates by forming soil macropores during nest construction. Cerdà and Jurgensen [81] showed that ant nests in intensively-managed orange orchard soil, characterized by a lack of vegetation cover, decreased water losses by the infiltration

of surface runoff into nest macropores. However, the unconsolidated soil mounds around ant nest entrances increased sediment yield and soil erosion rates, indicating that the presence of ants can increase soil erosion when rainfall intensity is greater than the infiltration capacity of the ant macropores. A similar experiment was conducted in an organic citrus orchard, where ants were more abundant than conventional management due to no-tillage practices, no pesticide use and the resulting vegetation cover. In the study of Cerdà et al. [82], it was shown that the application of sustainable practices to the soil (organic amendments, vegetation and litter cover), according to organic management, promoted high water infiltration rates, and ant activity had little effect on soil erosion rates.

Among different species of ants, leaf-cutting ants modify soil fertility through two mechanisms. First, the building, enlargement, and maintenance of nests ants affect soil structure, porosity and density. Second, leafcutters collect and concentrate vegetal material inside their nests to maintain their fungus culture, the food for most of the colony. The organic waste is very rich in nutrients. Consequently, plants inhabiting nest areas often show greater abundance, growth rate, foliar and root biomass and reproduction rates than plants outside nest areas. This positive effect on plants, due to the improvement of soil fertility might scale-up and affect the balance between trees and cover crops at landscape scale [43,44]. According to an agro-ecological view, the management of ants in fruit orchards requires knowledge of their positive and negative impacts on crops and pest species.

### 5.4. Termites

Termite is an endogenic exopterygotous insect that belongs to the order Isoptera and it is one of the numerous organisms that inhabit the soil. The abundance, composition and, hence, the impact on soil processes vary greatly depending on vegetation and land use [42,83]. Furthermore, it is known that termites are undoubtedly key soil organisms in tropical and subtropical soils. The most relevant literature on the effects of differential soil management on termite dynamics in orchard soils is summarized in Table 3.

**Table 3.** Effects of differential agricultural management on termite dynamics in orchard soils.

| Authors | Year | Fruit Orchard | Agricultural Management | Results | | Reference |
|---|---|---|---|---|---|---|
| Stansly et al. | 2001 | Citrus | Baiting with hexaflumuron bait | a) | Baiting systems provided longer-lasting control by eliminating or reducing termite activity | [83] |
| | | | | b) | Baiting with hexaflumuron appeared to be a viable alternative for managing subterranean termites in citrus and possibly other agricultural systems | |
| Coulibaly et al. | 2016 | Mango | Chronosequence of tree orchards (from young to old) with higher management in young compared to old (tillage; weeding; application of pesticides) | a) | The exploitation of the soil affected the trophic structure and the species richness of the fauna | [84] |
| | | | | b) | Species richness of termites was low in the young orchards | |
| | | | | c) | The diversity and the abundance of termites increased with the age of the orchards to reach a maximum in the old orchards | |
| | | | | d) | Production of signaling and energy-rich molecules that act as ecological mediators of biological engineering processes | |

In orchards, termites play an important role in the decomposition of litter on the ground, the regulation of soil structure, soil organic matter (SOM) and nutrient cycling, water dynamics, soil erosion, plant growth, and overall biodiversity [42]. Soil dwelling termites dig nests in the ground that have a significant impact on the soil environment. Activities of termites can result in the accumulation of organic matter and enrichment of nutrients and minerals in the soil [41]. Termites,

with a well-developed caste system, are highly successful, constituting up to 75% of the insect biomass and 10% of all terrestrial animal biomass in the tropics [42]. These insects have been considered as tropical analogs of earthworms since they process large amounts of litter. Three nutritional categories include wood-feeding species, plant- and humus-feeding species, and fungus growers. The last group lacks intestinal symbionts and depends upon cultured fungus for nutrition. Termites have an abundance of unique microbes living in their guts. Termite activities in orchard agrosystems affect the nutrient and organic matter dynamics and structure of the soil. Such changes in soil properties have a profound influence on the productivity of orchard agrosystems via carbon sequestration and nutrient cycling [42,84].

*5.5. Beetles*

Beetles (order Coleoptera) represent almost one-fourth of all described species in soil, and their species richness is associated with extreme morphological, ecological, and behavioral diversity. The order Coleoptera is classified into four suborders (Archostemata, Adephaga, Myxophaga and Polyphaga), 17 superfamilies, 168 families, approximately 370,000 identified species, and 3–5 million estimated species [7,85]. Most soil-dwelling beetle species are brown or black and, due to their high diversity of feeding habits, live in humus, leaf litter, decomposing roots and logs, other rotting organic matter, dung, carrion, and in the fruiting bodies of many types of fungi, where they significantly contribute to decomposition processes [7]. Soil beetles are of key importance in orchard soil dynamics. Indeed, many of them (particularly the species belonging to Carabidae, Leiodidae, Staphylinidae and Scarabaeidae) are well adapted to the soil environment and have special habits [36]. For instance, burying beetles bury carcasses of small vertebrates for feeding for their larvae. Some carrion beetles (family Silphidae) and some dung beetles (family Scarabaeidae) build nests in the soil. Many species of the family Staphylinidae live solely in caves while others are myrmecophiles (ant lovers) or termitophiles (termite lovers) [36].

Beetles are of key importance for orchard soil dynamics, as they are abundant and varied, and can exploit a wide diversity of soil–food sources [86,87]. They also regulate soil trophic nets as, from a functional point of view, they can be saprophagous, phytophagous or predators, and many species are predators of earthworms, collembolans, mites and nematodes [7]. In orchards, the presence of ground beetles on the soil surface over the entire growing season renders them highly sensitive to the long-term effects of plant protection products sprayed onto the trees [88]. Furthermore, Staphylinidae, one of the largest families of Coleoptera, includes species that are mostly predacious and have a role as biological regulators of crop pests [89]. Environmental conditions, wide-spectrum insecticides and soil management type have a significant effect on species richness and modify the dominance structures of the beetle communities [90]. In orchard soils, beetles are moderately to extremely inhibited by soil tillage, mostly depending on their size (the larger organisms are more likely to be reduced than the smaller organisms under tillage systems) [88,89]. For all these reasons, many species of Carabidae and Staphylinidae are being proposed as reliable indicators to assess soil health and they have been used as markers of environmental changes in orchard agro-systems. Carabid beetles are widespread insects that occur in many orchard agro-systems, and they are considered as potential control and frequently used to indicate habitat alteration [91]. Several studies show that their populations are favored by non-managed soils [92–95].

For their role of ecosystem engineers able to increase soil fertility, three beetle groups are particularly relevant in orchard soils: a) Scarabeidae (dung beetles), that dig sub-vertical galleries as anecic earthworms do, and whose presence is generally indicated by small mounds a few centimeters high on the soil surface, and b) Curculionidae and c) Melolonthinae, whose larvae (white grubs) may significantly affect crop production by feeding on living roots [7,36]. Because of the positive contribution of beetles to agricultural ecosystems, maintaining and developing their diversity through sustainable agricultural practices is recommended. The most relevant literature on the effects of differential soil management on beetle dynamics in orchard soils is summarized in Table 4.

**Table 4.** Effects of differential agricultural management on beetle dynamics in orchard soils.

| Authors | Year | Fruit Orchard | Agricultural Management | Results | Reference |
|---|---|---|---|---|---|
| Epstein et al. | 2001 | Apple | Conventionally managed orchards vs. no neural-active insecticides addition | Orchard blocks under conventionally managed regimes had significantly lower populations of ground beetles compared with orchard blocks managed without broad-spectrum insecticides | [88] |
| Miñarro and Dapena | 2003 | Cider-apple orchard | Six different groundcover management systems | The groundcover management affected the activity, density and diversity of ground beetles that contribute to the natural control of pests | [94] |
| Balog and Markó | 2007 | Apple, pear | Conventionally treated orchards and abandoned orchards | a) The activity and density of Staphylinidae were higher in pear, in sandy soil and abandoned orchards b) Types of soil treatment and cover crops played a role in forming the Staphylinidae communities | [95] |
| Balog et al. | 2009 | Apple, pear | Different environmental conditions (agricultural lowland environment; flooded areas; woodland areas of medium height mountains) | The cumulative effects of environmental conditions and soil modified the activity and density of Staphylinidae communities | [90] |
| Cotes et al. | 2009 | Olive | Organically and conventionally-managed soil | a) Maintenance of green covers decreased soil erosion risk, and traditional soil practices tended to diminish diversity and activity of the soil fauna b) Beetle assemblage under vegetable cover was more favorable c) Weeds offered advantage (hospitable temperature and moisture) to beetles, compared to bare soils | [92] |
| Honěk et al. | 2012 | Apple | Organic vs. conventional apple orchard | Positive role of rove beetles in the biological control of agricultural arthropod pests | [96] |
| Hedde et al. | 2015 | Apple | Conventionally-managed orchards vs. lower use of synthetic chemical compounds | The activity-density of ground beetle communities was solely influenced by season and species richness and by orchard management | [91] |
| Nietupski et al. | 2015 | Hazelnut | Different soil cultivation methods | Optimal soil tillage system promoted the presence of beetles in soil kept fallow with machines or chemicals, and in soil covered with manure | [97] |
| Marshall and Lynch | 2020 | Apple | No-till green cover vs. tilled soil | The density of some species of beetle (*Harpalus* spp.) increased with green cover | [86] |

## 6. Cases of Negative Effects of Macrofauna in Orchard Soils

Macrofauna is not always beneficial to orchard soils. Soil invertebrates can be invasive species able to affect soil carbon sequestration, soil fertility, and plant and animal health, resulting in economic and ecosystem change [31,34]. In some cases, the overall impact of macrofauna on microbial respiration in litter or soil-litter microcosms can be positive or negative. The most consistent effect of macrofauna in decomposing leaf litter is an increased rate of nitrogen mineralization, which results predominantly from interactions with microorganisms and not from excretion, and results in the accumulation of nitrates in soils in periods where plants do not take them up, thereby favoring their leaching [97].

Recently, Lubbers et al. [98,99] investigated the effect of earthworms on the greenhouse gas balance of soil, finding that earthworm presence stimulates carbon sequestration in soil aggregates, but at the same time, increases the combined cumulative emissions of $CO_2$ and $N_2O$ from a simulated no-tillage system to the same level as a simulated conventional tillage system. These findings might limit the potential benefits of sustainably-managed soils concerning global change as, although earthworms are largely beneficial to soil fertility, they could increase net soil greenhouse gas emissions. In fruit

orchards, ants and termites can have a negative impact, such as crop damage and fruit quality reduction during post-harvest [100], or the suppression of other beneficial arthropods due to interspecific competition [101]. Termites are highly devastating and polyphagous insect pests that cause damage to plants and crops, acting also as agricultural pests. They can attack plants at any stage of development, from the seed to the mature plant. Indeed, crops subjected to termite attack include 12 fruit plants (guava, coffee, citrus, banana, mango, papaya, grapes, mulberry, pineapple, almond, litchi and plum). To apply adequate soil management systems and the related pest control practices, it is necessary to consider the ecology and taxonomy of the local termite species [102,103]. Also, in the case of beetles, some species act as pests in fruit orchards. For example, adults of bronze beetles (*Eucolaspis* spp.) in organic apple orchards damage developing fruits, causing yield losses of up to 43% and high economic losses. Some strategies for the control of bronze beetle in apple orchards were developed using organically acceptable products, including pyrethrum and garlic [104,105].

Finally, although the intensification of agricultural practices in orchards usually has negative effects, Bucholz et al. [61] showed no clear response of macrofauna biodiversity in vineyards with permanently green cover crops, compared to vineyards periodically mechanically disturbed. Their results showed that the effects on soil biota of periodically-tilled vineyards seem to be not necessarily detrimental. The authors concluded that soil biota is influenced not only by management factors but also by the surrounding landscape structure and by plant biomass and soil quality

## 7. Conclusions

The analysis of the scientific literature highlighted the significant effects of soil macrofauna on soil quality and water dynamics in fruit orchards. The extensive networks of subterranean galleries and chambers built by ants and termites allow for better soil drainage and soil aeration [52,56,57]. On the other side, earthworms are responsible for soil mixing and transforming organic matter more readily assimilable by tree plants. Mean population densities of the main macrofaunal soil ecosystem engineers change across a chrono-sequence of soil maturation sites in fruit orchards at different ages (from incipient to young, mature and old weathered) [36]. In particular, earthworms and ants are present in higher abundance during the early stages of soil maturation, while termites population increase in more mature and old weathered soils. Macrofaunal species can be used as bioindicators of the quality of orchard soils. Biondication using macrofaunal organisms has been proved to be a useful tool for evaluating the state of conservation of an orchard soil and for discriminating between different soil management types [60]. Indeed, the orchard managed sustainably or organically register greater abundance, diversity, and richness of soil macrofaunal taxonomic groups.

In most cases, macrofauna biodiversity and abundance is affected by different fruit orchard management. Sustainable and organic farming is beneficial for soil biological activity of macrofauna [58,59]. This can be due to the higher accumulation of plant litter and organic matter in sustainably managed orchards. As a consequence of soil macrofauna abundance, higher production and reduced environmental pollution occur in orchard agrosystems, mainly due to the increase in soil quality and fertility. Conversely, conventional fruit production, due to the unavoidable lack of resources (particularly soil and water), and the heavy use of pesticides and mineral fertilizers is going to be economically and environmentally disadvantageous. For this reason, orchard management should be directed towards the principles of innovative, sustainable and conservative agriculture—the so-called "smart agriculture"—which rationally includes the existing and innovative agro-technological practices, such as no- or minimum soil tillage, onsite nutrient input and recycling, adequate irrigation and rational management of crop residues [106]. This integrated approach, aimed to keep fruit productions at high levels and cultivating in a lasting way, can render a wide range of benefits to farmers and the environment.

The world's soils are rapidly deteriorating due to soil erosion, nutrient depletion and other threats, but sustainable practices and technologies can reverse this trend. One key point from the new IPCC report [16] is that conventionally-managed soils erode more than 100 times faster than they form and

that land degradation represents "one of the biggest and most urgent challenges" that humanity faces. On this basis, better understanding the role of soil macrofauna in orchards could be a key factor for defining the best strategies of soil management in these agrosystems [2]. The role of macrofaunal organisms as colonizers, comminutors, and engineers within soils, together with their interactions with microorganisms, can contribute to the long-term sustainability of fruit orchards. Therefore, research about the taxonomy, structure and resilience of soil macrofaunal communities and their trophic net in differently-managed orchard soils can be of relevant importance [51], bearing in mind that the relationship between local changes (soil fauna/microorganisms) and global effects (soil quality/fertility, soil environmental importance, global change)—the so-called "local to global" concept—is particularly important in fruit groves, the products of which are a relevant source of income for many farmers and could deliver ecosystem services, if properly managed.

**Author Contributions:** Conceptualization, resources and data curation, A.S.; writing—review and editing, A.S., A.N.M., P.R.; supervision, A.N.M., P.R. All authors have read and agreed to the published version of the manuscript.

**Funding:** This research received no external funding.

**Conflicts of Interest:** The authors declare no conflict of interest.

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
