# Peer review of "Soil Macrofauna: A key Factor for Increasing Soil Fertility and Promoting Sustainable Soil Use in Fruit Orchard Agrosystems"

_agronomy, doi:10.3390/agronomy10040456_

Round 1

Reviewer 1 Report

The authors undertook an interesting study in an attempt to put together a series of bibliographic information in order to provide an overview of the soil macro-fauna and specifically as a key factor to promote the sustainable use of the soil and increase soil fertility in agrosystems orchards.

The authors carried out an accurate bibliographic survey and the piece manuscript is well written, in other parts little attention seems to have been paid to the use of the English language and to following the editorial rules of the Journal.

Furthermore, some parts of the manuscript are redundant and are repeated in the text in the paragraphs: 1. Introduction, 2. The scenario: Soil Organic matter in Orchard, 3. The actors: soil Biota in Agrosistem and 4. Soil Macrofauna abundance and activity are related to soil management.

Furthermore, many of the authors' statements in the text of the manuscript could flow into an important and more substantial discussion on the results produced by the careful research of the scientific literature on the subject. Therefore the manuscript is often quite heavy for the reader.

The conclusions that the authors present are not up to the work they have submitted. These are rather synthetic and do not shed light on the role of soil fauna in land management strategies and on restoring soil fertility and on the mitigation effect of climate change in orchards. The authors seem to gradually lose the direction indicated in the title of their work. I therefore suggest that the authors make the text more streamlined and improve the part relating to the conclusions, without losing sight of the aim of the work indicated in the title.

More specific comments:

Line 29 – remuve “fruit”;

Lines 79-84 – rewrite more clearly specifying the goal of the work;

Line  144 – elimina (Wagg et al., 2014) e segui le norme redazionali del giornale

Lines 165 – 168 questa affermazione necessita di riferimenti scientifici inserisci references;

Line 251 – elimina (Lavelle et al. 2016)  e segui le norme redazionali del giornale

Line 272 – cambia Orgiazzi and collegues in Orgiazzi et al.;

Line 284 – cambia Doles an collegues in Doles et al.;

Line 330 – cambia Paoletti and collegues in Paoletti et al.; 

Author Response

Comments and Suggestions for Authors

The authors undertook an interesting study in an attempt to put together a series of bibliographic information in order to provide an overview of the soil macro-fauna and specifically as a key factor to promote the sustainable use of the soil and increase soil fertility in agrosystems orchards.

The authors carried out an accurate bibliographic survey and the piece manuscript is well written, in other parts little attention seems to have been paid to the use of the English language and to following the editorial rules of the Journal.

Thank you. We edited the whole manuscript and corrected the English language.

Furthermore, some parts of the manuscript are redundant and are repeated in the text in the paragraphs: 1. Introduction, 2. The scenario: Soil Organic matter in Orchard, 3. The actors: soil Biota in Agrosistem and 4. Soil Macrofauna abundance and activity are related to soil management.

We tried to remove all the redundancies in the paragraphs you indicated, shortening this section.

Furthermore, many of the authors' statements in the text of the manuscript could flow into an important and more substantial discussion on the results produced by the careful research of the scientific literature on the subject. Therefore the manuscript is often quite heavy for the reader.

We corrected the manuscript, as you can see in the tracking-mode document, focusing more on the results produced. The subject has been generalized (not only Mediterranean orchards) and the conclusion part improved, including the discussion of the main data resulting from the review of the papers.

The conclusions that the authors present are not up to the work they have submitted. These are rather synthetic and do not shed light on the role of soil fauna in land management strategies and on restoring soil fertility and on the mitigation effect of climate change in orchards.

The conclusions have been completely rewritten and focused on the aims of the review (as written in the final part of the introduction), that is soil quality and fertility. The parts regarding climate change were reduced, as they were not the main subject of the paper.

The authors seem to gradually lose the direction indicated in the title of their work. I therefore suggest that the authors make the text more streamlined and improve the part relating to the conclusions, without losing sight of the aim of the work indicated in the title.

Please see the comment above.

More specific comments

Line 29 – remuve “fruit”;

Done

Lines 79-84 – rewrite more clearly specifying the goal of the work;

Done (please see the point above)

Line 144 – elimina (Wagg et al., 2014) e segui le norme redazionali del giornale

Done

Lines 165 – 168 questa affermazione necessita di riferimenti scientifici inserisci references;

Done

Line 251 – elimina (Lavelle et al. 2016) e segui le norme redazionali del giornale

Done

Line 272 – cambia Orgiazzi and collegues in Orgiazzi et al.;

Done

Line 284 – cambia Doles an collegues in Doles et al.;

Done

Line 330 – cambia Paoletti and collegues in Paoletti et al.; 

Done

Reviewer 2 Report

Review “Soil macrofauna: a key factor for promoting sustainable soil use and increasing soil fertility in fruit orchard agroecosystems”

Major changes:
Please give some description of the distribution of soil fauna with depth. To what degree does the tillage of orchards impact the soil biota occupying different depths in the soil profile?
Similar comment for Fig. 3: at what depths does the density of roots most change
Overall, please include the effect of different chemical and physical fractions of SOM on soil foodwebs.
Please also discuss differences between particulate organic matter (POM) and mineral-associated organic matter (MOM). What soil types exist in the Mediterranean region considered in the review? How would the texture and mineral composition of soils in the region effect fractions of SOM available to soil biota?

Fig. 1 Please cite the source of the presented data.
Table 3. Are all of these crops grown in the region subject to this review (Mediterranean agro-systems)?

Minor changes:
L42 “…all of which ARE essential…”
L104. replace “in the next future” with “in the future”
L. 107. “…all of which ARE essential…”
L.144 cite number, not reference to be consistent
L251 cite number, not reference to be consistent
L467. “…able TO…”

Author Response

Major changes

Please give some description of the distribution of soil fauna with depth. To what degree does the tillage of orchards impact the soil biota occupying different depths in the soil profile?
Similar comment for Fig. 3: at what depths does the density of roots most change

Thank you. Although it is not the main subject of the review paper, we also consider the aspect of soil depth as important and relevant. For this reason, we added a paragraph at the end of paragraph 4. We highlighted the agronomic importance of the topsoil and commented, in general, the functional types of macrofaunal organisms living at different soil depths. We also added a new figure (Fig. 4) for describing these aspects.

Overall, please include the effect of different chemical and physical fractions of SOM on soil foodwebs. Please also discuss the differences between particulate organic matter (POM) and mineral-associated organic matter (MOM). What soil types exist in the Mediterranean region considered in the review? How would the texture and mineral composition of soils in the region effect fractions of SOM available to soil biota?

We agree and, considering that in this review we put a big emphasis on SOM, a whole paragraph has been added at the end of paragraph 2, describing the differences between POM and MOM and their relationships with macrofauna.

Fig. 1 Please cite the source of the presented data.

Done

Table 3. Are all of these crops grown in the region subject to this review (Mediterranean agro-systems)?

The focus of this review paper are orchards in general and not only Mediterranean agro-systems, so, in this revised version, a more general cut has been given.

Minor changes

L42 “…all of which ARE essential…”

Done

L104. replace “in the next future” with “in the future”

Done

  1. 107. “…all of which ARE essential…”

Done

L.144 cite number, not reference to be consistent

Done

L251 cite number, not reference to be consistent

Done

L467. “…able TO…”

Done

Reviewer 3 Report

This review is well presented and fills a gap in the perennial cropping area.  The english grammar needs to be improved, so the authors should ensure that they enlist the assistance of someone with English as a first language.

In section 5 there is a focus on earthworms, ants and termites, but there are other soil macrofauna that play an important role in soil health (eg. some beetles), so these should be discussed as well.  Additionally meso-fauna such as collembola can be used as soil health indicators, so perhaps the authors should consider the inclusion of meso-fauna in the review.

Author Response

Comments and Suggestions for Authors

This review is well presented and fills a gap in the perennial cropping area. The english grammar needs to be improved, so the authors should ensure that they enlist the assistance of someone with English as a first language.

Thank you. We edited the whole manuscript and improved the English grammar.

In section 5 there is a focus on earthworms, ants and termites, but there are other soil macrofauna that play an important role in soil health (eg. some beetles), so these should be discussed as well.

We agree and, according to your suggestion, we added a new paragraph (5.4) on the importance of beetles in orchard agro-systems. Some of the most relevant references [85-92] have been commented. A new table (Table 4) was added.

Additionally, meso-fauna such as collembola can be used as soil health indicators, so perhaps the authors should consider the inclusion of meso-fauna in the review.

We understand the importance of collembola but, on the other side, they belong to mesofauna, that is not the topic of this review. Therefore, we added some references and cited their interaction with macrofauna and microorganisms throughout the text, without adding a separate paragraph or other sections. Indeed, we believe the review is long enough and focused on macrofauna (as written in the title), and we wouldn't want to overload it with too many topics.

Round 2

Reviewer 1 Report

Dear Authors,

the new version of the manuscript, in which you made the suggestions proposed by the reviewers, appears significantly improved. At the same time, however, I must invite you to elaborate better on the paragraph concerning Beetles. As you have well written, Beetles represent almost one-fourth of all described species and their species richness is associated with extreme morphological, ecological, and behavioral diversity.

The paragraph on beetles is poorly approached, the reader would expect a more substantial analysis, considering that you have made a good description for earthworms, ants and termites. There are numerous recent studies on the activity that Coleoptera carry out in the soil, insert them.

In the caption of Figure 1 since the source refers to a website.
It is necessary to include in the references the website from which the image was copied and the month in which the website was visited.

The manuscript presented in this form is not yet ripe for publication in the Agronomy Journal.

Author Response

Firstly, thank you for your further revision of the manuscript. We tried to make the changes you requested, hoping that the paper has ben improved.

The new version of the manuscript, in which you made the suggestions proposed by the reviewers, appears significantly improved. The manuscript presented in this form is not yet ripe for publication in the Agronomy Journal.

Thank you.

At the same time, however, I must invite you to elaborate better on the paragraph concerning Beetles. As you have well written, Beetles represent almost one-fourth of all described species and their species richness is associated with extreme morphological, ecological, and behavioral diversity. The paragraph on beetles is poorly approached, the reader would expect a more substantial analysis, considering that you have made a good description for earthworms, ants, and termites. There are numerous recent studies on the activity that Coleoptera carry out in the soil, insert them.

We have completely rearranged the paragraph on beetles, adding more details, focusing on their importance in orchard agro-systems and commenting on the more relevant and recent studies on Coleoptera (5 new references). Therefore, we hope that the paragraph has been improved. However, we also paid attention to not unbalance paragraph 5, so we narrowed the discussion on beetles to the most important aspects, particularly on orchard soils.

In the caption of Figure 1 since the source refers to a website. It is necessary to include in the references the website from which the image was copied and the month in which the website was visited.

It has been done (new reference number: [17]).

Kind regards